# Prevalence and patterns of multimorbidity in the Jamaican population: A comparative analysis of latent variable models

**Leslie S. Craig**[1], **David R. Hotchkiss**[2], **Katherine P. Theall**[2], **Colette Cunningham-Myrie**[3]*, **Julie H. Hernandez**[4], **Jeanette Gustat**[5]

1 Department of Medicine, School of Medicine, Tulane University, New Orleans, Louisiana, United States of America, 2 Department of Global Community Health and Behavioral Sciences, School of Public Health and Tropical Medicine, Tulane University, New Orleans, Louisiana, United States of America, 3 Department of Community Health and Psychiatry, University of the West Indies, Mona, Jamaica, 4 Department of Health Policy and Management, School of Public Health and Tropical Medicine, Tulane University, New Orleans, Louisiana, United States of America, 5 Department of Epidemiology, School of Public Health and Tropical Medicine, Tulane University, New Orleans, Louisiana, United States of America

* colette.cunninghammyrie@uwimona.edu.jm

**Data Availability Statement:** All relevant data are within the manuscript and its Supporting Information files.

## Abstract

### Background

Evidence suggests that the single-disease paradigm does not accurately reflect the individual experience, with increasing prevalence of chronic disease multimorbidity, and subtle yet important differences in types of co-occurring diseases. Knowledge of multimorbidity patterns can aid clarification of individual-level burden and needs, to inform prevention and treatment strategies. This study aimed to estimate the prevalence of multimorbidity in Jamaica, identify population subgroups with similar and distinct disease profiles, and examine consistency in patterns identified across statistical techniques.

### Methods

Latent class analysis (LCA) was used to examine multimorbidity patterns in a sample of 2,551 respondents aged 15–74 years, based on data from the nationally representative Jamaica Health and Lifestyle Survey 2007/2008 and self-reported presence/absence of 11 chronic conditions. Secondary analyses compared results with patterns identified using exploratory factor analysis (EFA).

### Results

Nearly one-quarter of the sample (24.1%) were multimorbid (i.e. had ≥2 diseases), with significantly higher burden in females compared to males (31.6% vs. 16.1%; *p*<0.001). LCA revealed four distinct classes, including a predominant *Relatively Healthy* class, comprising 52.7% of the sample, with little to no morbidity. The remaining three classes were characterized by varying degrees and patterns of multimorbidity and labelled *Metabolic* (30.9%), *Vascular-Inflammatory* (12.2%), and *Respiratory* (4.2%). Four diseases determined using physical assessments (obesity, hypertension, diabetes, hypercholesterolemia) were

**Funding:** The author(s) received no specific funding for this work.

**Competing interests:** The authors have declared that no competing interests exist.

primary contributors to multimorbidity patterns overall. EFA identified three patterns described as "Vascular" (hypertension, obesity, hypercholesterolemia, diabetes, stroke); "Respiratory" (asthma, COPD); and "Cardio-Mental-Articular" (cardiovascular disease, arthritis, mental disorders).

## Conclusion

This first study of multimorbidity in the Caribbean has revealed a high burden of co-existing conditions in the Jamaican population, that is predominantly borne by females. Consistency across methods supports the validity of patterns identified. Future research into the causes and consequences of multimorbidity patterns can guide development of clinical and public health strategies that allow for targeted prevention and intervention.

## Introduction

Non-communicable diseases (NCDs) have been established as the primary cause of morbidity, with a considerable attendant premature mortality burden that disproportionately impacts poor, vulnerable and socio-economically disadvantaged populations within low- and middle-income countries (LMICs) [1–3]. Adding to the social, financial and physical burdens associated with management of NCDs is the predominant single-morbidity approach of clinical care guidelines, despite evidence that these diseases seldom occur in isolation, with an increasing proportion of persons experiencing multiple coexisting chronic diseases or multimorbidity (i.e. the co-occurrence of two or more diseases) [4–8].

Although a growing body of literature is available on the patterns and clusters of diseases, multimorbidity remains a complex phenomenon, with a vast variety of potential disease combinations that make it difficult to analyze [9]. Moreover, in the absence of an established "gold standard" measurement, considerable variation exists in the application of statistical methods to studies of this phenomenon [4,9–11]. Previous studies have typically relied on simple disease counts to specify whether a person has two or more conditions from a pre-defined list [4,10,12]. Exploratory factor analysis (EFA) and traditional cluster analysis techniques have also emerged as commonly used methods [7], with latent class analysis (LCA) being increasingly applied to studies of multimorbidity patterns [9,11,13–17]. Yet, despite recognized—and increasing—methodological diversity, few studies have endeavored to increase the reliability of findings through comparison of statistical techniques.

To date, only two studies have compared multimorbidity patterns identified using different analytic approaches. One study of 408,994 patients aged 45–64 years in Catalonia, Spain extracted diagnoses from electronic health records, using 263 disease blocks of the International Classification of Diseases version 10 (ICD-10), to compare patterns identified via hierarchical cluster analysis and EFA methods [18]. Authors concluded that while disease groupings from the two analytic methods did not always match exactly, there was some consistency in multimorbidity patterns [18]. The other study used self-reported data on 10 NCDs from a cross-sectional sample of 4,574 Australian adults, 50 years and older, finding consistency in results across four methods (i.e. commonly occurring pairs and triplets of comorbid diseases; cluster analysis of diseases; principal component analysis; LCA) that was suggestive of the co-occurrence of diseases beyond chance [11]. Notably, despite evidence of variation in the burden of individual NCDs across population subgroups [1], neither study examined sex

differences in multimorbidity patterns. However, both studies did emphasize the need to strengthen the evidence base on multimorbidity prevalence and patterns, to better inform disease management and healthcare delivery [11,18]. International organizations, such as the World Health Organization (WHO), the European Forum for Primary Care and the National Institute for Health and Clinical Excellence (NICE), similarly echo this sentiment noting that knowledge of multimorbidity patterns in a given population is an important first step towards generating an evidence base for actual clinical practice, with significant implications for patient-oriented prevention, diagnosis, treatment, and prognosis [7,19,20].

Throughout WHO regions worldwide, the burden of NCDs is purportedly highest in the Americas, with higher rates among people in the English-speaking Caribbean nations [2,21,22]. Within the Caribbean community, Jamaica has been conducting numerous comprehensive national health surveys, including the Jamaica Health and Lifestyle Surveys 2000/2001 (JHLS-I) and 2007/2008 (JHLS-II) [23–26], providing a well-established evidence base of a severe burden of individual NCDs, that is predominantly borne by females [23–27]. However, no investigation of multimorbidity prevalence or patterns has yet been undertaken for Jamaica, or the larger Caribbean region.

This study aims to address this research gap via secondary analysis of the JHLS-II dataset. First, LCA was used to describe the prevalence of multimorbidity in the Jamaican population, identify classes of individuals with distinct multimorbidity patterns and examine whether these patterns were similar across sex. Then, to assess the validity and reliability of multimorbidity profiles identified via LCA, EFA was used as a robustness check to compare consistency (or variation) in patterns identified across the two latent modelling techniques. Results from this study will provide nuanced insight into the burden and distribution of co-occurring conditions in the Jamaican population, to inform more targeted prevention and management strategies.

## Methods

### Sample

The JHLS-II is a nationally representative study that was coordinated at the Epidemiology Research Unit (ERU) of the Tropical Medicine Research Institute (TMRI), the University of the West Indies, Mona, recruiting a sample of 2,848 Jamaicans, 15–74 years of age over a four-month period spanning from November 2007 and March 2008, via a multi-stage cluster sampling design [26,27]. In brief, participant recruitment was based on a random selection of clusters (or enumeration districts) proportionate to the size of the population within the 14 parishes of Jamaica [26]. Enumeration districts were determined by the Statistical Institute of Jamaica. Within each cluster, a random starting point was chosen and every 10th household systematically identified, with a single individual from each household being invited to participate [26]. An interviewer-administered questionnaire was used to obtain self-reported information on demographic characteristics, medical history and health behaviors. Physical (i.e. height, weight, waist circumference) and biological (i.e. blood pressure, blood glucose, total cholesterol) measurements were made in accordance with standardized protocols [26,27]. Low non-response rate (1.7%) and maintenance of high inter- and intra-observer reliabilities throughout the survey were indicators of good data quality [26]. Further details of the survey design, sampling procedures and data collection methods are provided in the technical report [26].

### Measures

Indicators of multimorbidity were limited to those NCDs with the greatest burden in the population (i.e. prevalence greater than or equal to 1% in each sex). Following guidance from the

2011 systematic review on multimorbidity measurement by Diederichs and colleagues that related diseases be combined [10], cardiovascular disease (i.e. heart disease, myocardial infarction, and circulation problems) and mental health disorders (i.e. depression, anxiety, psychosis, and other mental health problems) were grouped together to enhance data quality. Self-reported diagnosis of bronchitis/pneumonia was used as a proxy indicator of chronic obstructive pulmonary disease (COPD). The final list of 11 conditions included hypertension, obesity, hypercholesterolemia, diabetes, asthma, arthritis, cardiovascular disease, mental health disorders, COPD, stroke, and glaucoma.

Presence or absence of these final 11 conditions was largely based on self-report, with the exception of four diseases (obesity; hypertension; diabetes; hypercholesterolemia) where physical assessments were available and used alone, or in combination with self-reports, to increase measurement validity and reliability. Specifically, objective measurements of height and weight were used to determine obesity status (body mass index, BMI, $\geq$30 kg/m$^2$), in accordance with WHO guidelines [1]. Diabetes was defined as having a fasting plasma glucose value $\geq$7.0 mmol/L (126 mg/dl) or being on medication for raised blood glucose [1]. Hypertension was defined as systolic blood pressure $\geq$140 mmHg and/or diastolic blood pressure $\geq$90 mmHg or using medication to lower blood pressure [1]. Hypercholesterolemia was defined as total cholesterol levels of 5.2 mmol/l or higher or self-reported use of medications to control blood cholesterol [1].

Multimorbidity was defined as having two or more of the final list of 11 NCDs. A 2012 systematic review by Fortin and colleagues advised inclusion of at least 2 operational definitions of multimorbidity: (1) presence of two or more diseases; and (2) presence of three or more diseases; noting that the latter definition may be more meaningful for clinicians given that a simple count of 2 or more diseases is less discriminating [4]. Accordingly, descriptive analyses also use the latter definition, to allow for identification of individuals with higher needs and greater disease burden [4].

## Ethics

Ethical approval of the JHLS-II survey instruments and procedures was granted by the Ministry of Health, Jamaica and the University of the West Indies.

## Statistical approach

Analyses were restricted to participants with non-missing information on the 11 NCD multimorbidity indicators. Of the 2,848 respondents who completed the survey, 311 (10.9%) were missing information on one or more of these indicators. There were no statistically significant differences between those with complete and those with missing information on the basis on sex, age or region of residence (all $p$>0.05). The final analytic sample of 2,551 respondents included 790 males and 1,761 females.

Descriptive statistics were calculated for the overall sample and each sex group, to determine the prevalence of morbidity (from individual NCDs) and multimorbidity. Means with 95% confidence intervals (95% CIs) (for continuous variables), and proportions (for categorial variables) were computed and compared using the Mann-Whitney U test and the Pearson's chi-squared ($\chi^2$) test, respectively, to examine differences across sex. All analyses were weighted to account for sampling design and non-response as well as differences in the age-sex distribution of the study sample compared to the Jamaican population. Base sampling weights reflected the product of the inverse of the probability of selecting a household and the inverse of the probability of selecting a primary sampling unit, adjusted for non-response. Post-stratification weights were calculated as the number of persons in the Jamaican population between

the ages of 15–74 years, represented by each individual in the sample within 5-year age-sex categories.

**Latent class analysis (LCA).** *Identification of the baseline model.* LCA was used to identify discrete, mutually exclusive classes of individuals with distinct multimorbidity patterns, based on the presence or absence of the final list of 11 NCD indicators. In order to identify an optimal baseline model, a sequence of LCA models was examined beginning with a single-class model and adding classes in a stepwise fashion until model fit no longer significantly improved. Models with 1 through 6 classes were fit to the data, with final model selection based on a balance of parsimony, substantive consideration of each model and comparison of a range of model fit indices. To ensure that the global maximum (rather than local maximum) was identified, an iterative maximum likelihood estimate was used, with a minimum of 200 'random' sets of starting values [28,29]. The number of random sets was increased as needed to achieve model identification (i.e. one frequently occurring, dominant solution where the log-likelihood and parameter estimates are replicated) [28,29].

Several indices were used to guide model selection, including the likelihood-ratio $G^2$ statistic, the Akaike Information Criteria (AIC), the Bayesian Information Criteria (BIC) and the adjusted BIC [29,30]. The likelihood-ratio $G^2$ statistic (and parametric bootstrap likelihood ratio test) were used to test the null hypothesis that the specified LCA model fit the data (i.e. a significant p-value indicated that the null model was too restrictive) [28,29]. Lower values on the information criteria were indicative of a more optimal balance between model fit and parsimony [28,29]; greatest weight was given to the AIC following evidence from simulation studies of serious underfitting of the BIC, particularly with smaller samples and more unequal class sizes [31,32]. Substantive interpretability was considered via inspection of probability plots to ensure that resultant solutions were distinguishable, non-trivial in size, and meaningful [29]. The prevalence of each latent class was calculated as the average across participant-specific class membership probabilities [29]. Once the baseline model had been selected, participants were assigned to their best fit class based on their maximum posterior probability and the mean posterior probability of each latent class calculated as an indicator of classification certainty [33]. Mean posterior probabilities above 70% indicated optimal fit [16].

Given the potential for obesity to have a double impact, as a *risk factor* for individual NCDs and as a *disease* requiring intervention, sensitivity analyses explored patterns of multimorbidity based on 10 of the 11 NCD indicators listed above (i.e. excluding obesity).

*Testing measurement invariance across sex.* Following guidelines by Lanza et al (2007) which recommend that analyses begin by first fitting a baseline model with no grouping variable [29], sex was added as a grouping variable after the baseline model had been selected, to test the hypothesis that multimorbidity patterns vary across sex. To test measurement invariance empirically, the model was run with all parameters freely estimated and again with item-response probabilities constrained equal across groups. The difference in the $G^2$ statistic between the two models was compared to the chi-square distribution for the difference in the models' degrees of freedom, and a significant p-value indicated different patterns across groups [28,29].

**Exploratory factor analysis (EFA).** EFA was used as a robustness check to examine similarities and/or differences in multimorbidity patterns identified using this latent modelling technique and the latent class approach. Consistent with the definition of multimorbidity as the coexistence of two or more diseases, an identified factor needed consist of at least two diseases to qualify as a multimorbidity pattern. Based on examples used in previous studies [6,34], along with the recommendations from systematic reviews [4,10], the following criteria were applied during EFA: only those NCDs with a prevalence ≥1% in each sex were included; data on NCDs were coded in binary form and tetra-choric correlation matrices used, owing to

the dichotomous nature of the NCD variables; and the principal components extraction method applied. The principal components extraction method allowed for determination of the number of factors to retain, in combination with eigenvalues >1 and scree plots to visually guide selection. Finally, owing to correlations between NCDs, the oblique rotation method was used to evaluate the factor solution and facilitate interpretation of factor loadings. Factor loadings >0.3 were taken as the minimum acceptable value for a significant correlation in the identification of diseases comprising each multimorbidity pattern. The Kaiser-Meyer-Olkin (KMO) statistic was used as a measure of sample adequacy [35,36].

All statistical analyses were carried out via Stata v.15 software, using the LCA Stata Plugin [37] and LCA Bootstrap Stata macro [38] as needed, with statistical significance indicated by a *p*-value <0.05.

## Results

### Sample description

Of the 11 NCD indicators included in the LCA, two diseases had an overall prevalence of about 25.0%, four had a prevalence between 5.0%–12.0%, while the remaining five had lower prevalence, typically under 5.0% (Fig 1). Among this sample of the Jamaican population, hypertension was the most prevalent NCD (25.3%), followed by obesity (25.2%), hypercholesterolemia (11.5%), diabetes (7.9%) and asthma (6.9%). About one third (30.6%) of the sample reported only one NCD while nearly one-quarter (24.1%) reported multimorbidity (i.e. two or more diseases). When the more discriminating definition of multimorbidity was applied, approximately 1 in every 10 participants (10.2%) reported at least 3 NCDs.

The multimorbidity burden was significantly greater in females (p<0.001), regardless of the definition used. In addition, there were statistically significant sex differences in the prevalence

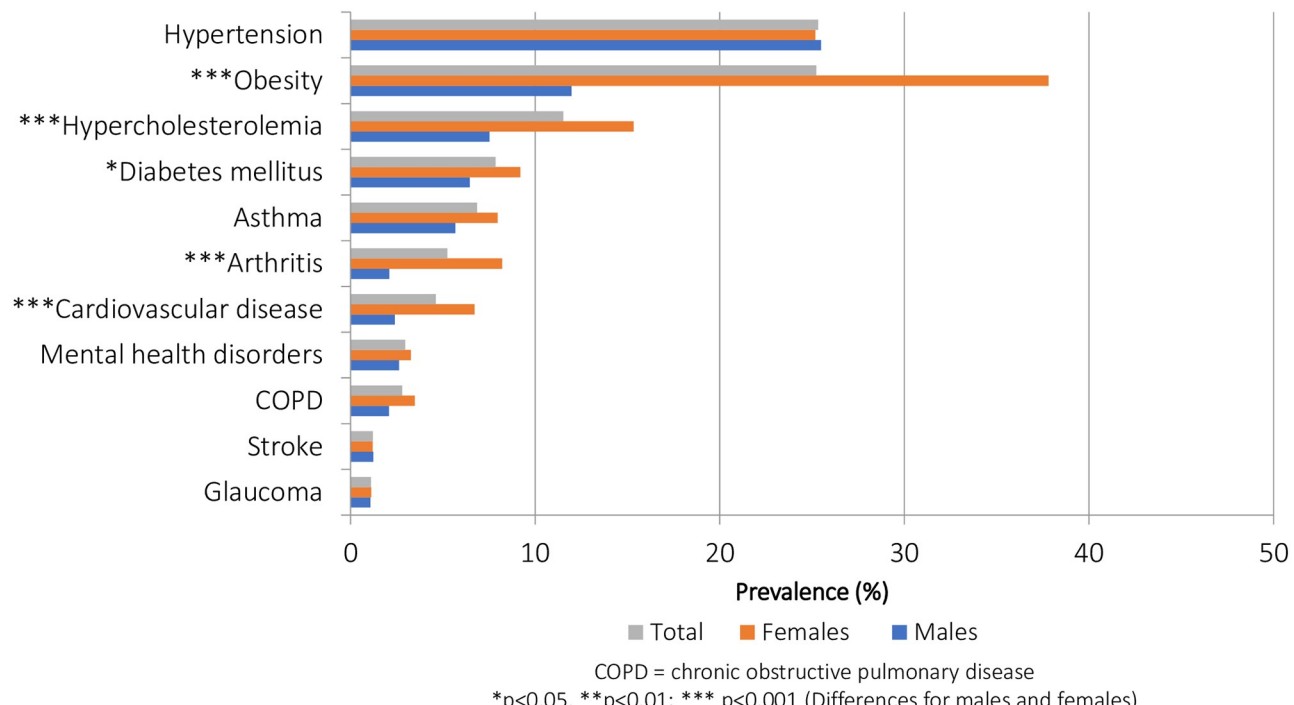

**Fig 1. Prevalence of non-communicable diseases (NCDs), by sex (JHLS-II data, 2007/2008; N = 2,551).**

of obesity (females:37.8% vs. males:12.0%; $p<0.001$), hypercholesterolemia (females:15.3% vs. males:7.5%; $p<0.001$), diabetes (females:9.2% vs. males:6.5%; $p<0.05$), arthritis (females:8.2% vs. males:2.1%; $p<0.001$), and cardiovascular disease (females:6.7% vs. males:2.4%; $p<0.001$), with the burden in females often 2 to 3 times as high as that in males. On average, females reported 1.2 NCDs (95% CI: 1.1–1.3) while males reported 0.7 diseases (95% CI: 0.6–0.7) ($p<0.001$).

## LCA baseline model

The LCA model fit results are summarized in Table 1.

The $G^2$ statistic, AIC and adjusted BIC consistently decreased up until the four-class model while the BIC reached a minimum in the two-class model. Neither models with five nor six classes were well identified—meaning that even after increasing the random starts so that the estimation procedure went through a maximum set of 400 iterations, neither model converged on the same solution the majority of the time. Notably, while the adjusted BIC indicated that the 4-class model was the best fit model, it suggested relatively little difference between this and the three-class model (adjusted $BIC_{4class}$ = 686.4 vs. adjusted $BIC_{3class}$ = 688.0; difference = 1.6). Nonetheless, the four-class model was better identified than the three-class model, with the maximum likelihood estimate converging on the same solution 97.5% of the time (compared to 55.0% of the time for the three-class solution). The four-class solution's entropy score (0.6) indicated greater precision in class prediction (compared to the three-class solution) and, upon examination, allowed for meaningful interpretation of latent classes. Results of the parametric bootstrap likelihood ratio tests (Table 2) further supported this decision, finding statistically significant differences for all except the four-class null model and the alternative five-class model ($p = 0.33$), indicating that the four-class model was the optimum baseline model.

Latent class prevalences and item-response probabilities (i.e. the estimated probability of reporting a particular NCD, given membership in a particular latent class) for the four-class model are graphed in Fig 2.

**Table 1. Summary of information for selecting number of multimorbidity latent classes (JHLS-II data, 2007/2008; N = 2,551).**

| Number of Latent Classes | $G^2$ | df | AIC | BIC | Adjusted BIC | log-likelihood | Entropy |
|---|---|---|---|---|---|---|---|
| 1 | 1318.4 | 2036 | 1340.4 | 1404.7 | 1369.8 | -8186.5 | 1.0 |
| 2 | 597.8 | 2024 | 643.8 | 778.2 | 705.1 | -7826.1 | 0.7 |
| 3 | 524.6 | 2012 | 594.6 | 799.2 | 688.0 | -7789.6 | 0.5 |
| 4 | 467.0 | 2000 | 561.0 | 835.7 | 686.4 | -7760.8 | 0.6 |
| 5 | Not well identified | | | | | | |
| 6 | Not well identified | | | | | | |

df = degrees of freedom; AIC = Akaike Information Criterion, BIC = Bayesian Information Criterion.

**Table 2. Model comparison for selecting the number of multimorbidity latent classes (JHLS-II data, 2007/2008; N = 2,551).**

| Null model | vs. | Alternative model | p-value |
|---|---|---|---|
| 1-class | | 2-class | 0.01 |
| 2-class | | 3-class | 0.01 |
| 3-class | | 4-class | 0.01 |
| 4-class | | 5-class | 0.33 |

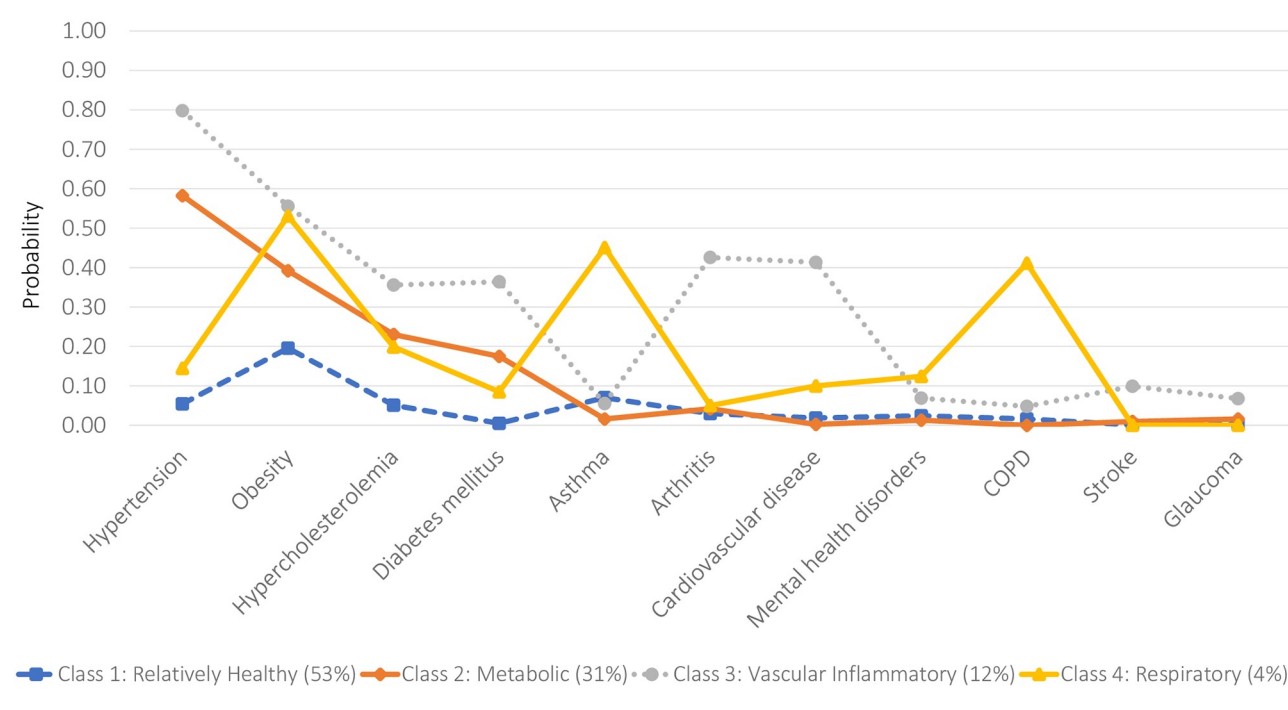

**Fig 2. Item-response probabilities for the four-class model (JHLS-II data, 2007/2008; N = 2,551).**

*Class 1* was labelled *Relatively Healthy* as it was characterized by individuals with low probabilities of all 11 NCDs. The majority of sample respondents (52.7%) were classified into this relatively healthy class. The mean number of NCDs was 0.4. *Class 2* was characterized by individuals with a high probability of hypertension and obesity, and somewhat moderate probability of hypercholesterolemia. This class was labelled *Metabolic* and comprised 30.9% of the sample. The mean number of NCDs in this *Metabolic* class was 1.6. Approximately one in five (19.8%) participants in this class had at least three NCDs. *Class 3* was characterized by individuals with a very high probability of hypertension, obesity, hypercholesteremia and diabetes. Specifically, members of Class 3 had a higher probability of these four NCDs than all other classes. Class 3 was also marked by an increased likelihood of arthritis and cardiovascular disease. This class was labelled *Vascular-Inflammatory* and comprised 12.2% of the sample. The mean number of NCDs was 3.4. The final class, *Class 4,* was characterized by individuals with the highest probability of asthma and COPD and was accordingly labelled *Respiratory*. This was the smallest of all classes, comprising 4.2% of the sample. The mean number of NCDs for the *Respiratory* class was 2.9.

The mean posterior probabilities for all four classes exceeded 0.7 (0.8 for the *Relatively Healthy* class; 0.8 for the *Metabolic* class; 0.9 for the *Vascular-Inflammatory* class; and 0.8 for the *Respiratory* class) suggesting optimal classification.

Sensitivity analyses exploring multimorbidity patterns using only 10 NCDs (i.e. excluding obesity), corroborated findings from the original baseline model with 11 NCD indicators. Specifically, LCA model fit statistics and results of the parametric bootstrap likelihood ratio test (S1 Table) all pointed to the 4-class model as the optimal baseline solution. Further, results of the four-class solution suggested that the latent classes were similarly characterized as *Relatively Healthy*, *Metabolic*, *Vascular-Inflammatory* and *Respiratory* based on the item-response

**Table 3. Fit statistics for test of measurement invariance across sex (JHLS-II data, 2007/2008; N = 2,551).**

| | $G^2$ | df | AIC | BIC | Adjusted BIC | log-likelihood |
|---|---|---|---|---|---|---|
| Model 1: Item-response probabilities free to vary across genders | 595.1 | 4001 | 783.1 | 1332.5 | 1033.8 | -7580.7 |
| Model 2: Item-response probabilities constrained equal across genders | 725.7 | 4045 | 825.7 | 1118.0 | 959.1 | -7646.0 |

$G^2(_2)$—$G^2(_1)$ = 130.6, df = 44, $p$ <0.01

df = degrees of freedom; AIC = Akaike Information Criterion, BIC = Bayesian Information Criterion.

probabilities; although a larger proportion of the sample was classified as being *Relatively Healthy* (with an almost negligible probability of reporting any NCD) and a smaller proportion was classified as having multimorbidity (see S1 Fig).

## Measurement invariance

To test measurement invariance across sex, the four-class solution was estimated, first using a model with all parameters free to vary across groups and, second, in a model with the item-response probabilities constrained equal across groups (Table 3).

The $G^2$ difference test was significant ($G^2(_2)$—$G^2(_1)$ = 130.6, df = 44, $p$ <0.01), suggesting that measurement invariance across sex did not hold and that the two groups should be modelled separately. Accordingly, a series of models were fit to individual male and female datasets to further investigate the driver of differences with the identified four-class latent structure. Table 4 shows the model fit statistics for the male and female subsamples, separately.

For both the male and female cohort, the AIC suggested a 4-class model while the BIC suggested the 2-class model. However, for the male cohort, the adjusted BIC reached a minimum with the 3-class model while, for the female cohort, it did so with the 2-class model. Based on the AIC for each subsample, in addition to examination of the distribution of item-response probabilities across all solutions, the 4-class model appeared to provide the best interpretability in each case. Parametric bootstrap analyses further supported this conclusion indicating that,

**Table 4. Summary of information for selecting the number of multimorbidity latent classes for male and female subsamples (JHLS-II data, 2007/2008; N = 2,551).**

**Males only (N = 790)**

| No. of Latent Classes | $G^{2*}$ | df | AIC | BIC | Adjusted BIC | log-likelihood | Entropy |
|---|---|---|---|---|---|---|---|
| 1 | 445.9 | 2036 | 467.9 | 519.3 | 484.4 | -1952.8 | 1.0 |
| 2 | 279.3 | 2024 | 325.3 | 432.8 | 359.7 | -1869.5 | 0.6 |
| 3 | 221.0 | 2012 | 291.0 | 454.5 | 343.3 | -1840.3 | 0.8 |
| 4 | 187.6 | 2000 | 281.6 | 501.2 | 351.9 | -1823.7 | 0.8 |
| 5 | Not well identified | | | | | | |
| 6 | Not well identified | | | | | | |

**Females only (N = 1,761)**

| No. of Latent Classes | $G^{2*}$ | df | AIC | BIC | Adjusted BIC | log-likelihood | Entropy |
|---|---|---|---|---|---|---|---|
| 1 | 1028.5 | 2036 | 1050.5 | 1110.8 | 1075.8 | -6081.9 | 1.0 |
| 2 | 490.5 | 2024 | 536.5 | 662.4 | 589.3 | -5812.9 | 0.6 |
| 3 | 441.4 | 2012 | 511.4 | 702.9 | 591.8 | -5788.4 | 0.6 |
| 4 | 407.4 | 2000 | 501.4 | 758.7 | 609.4 | -5771.4 | 0.6 |
| 5 | Not well identified | | | | | | |
| 6 | Not well identified | | | | | | |

df = degrees of freedom; AIC = Akaike Information Criterion, BIC = Bayesian Information Criterion

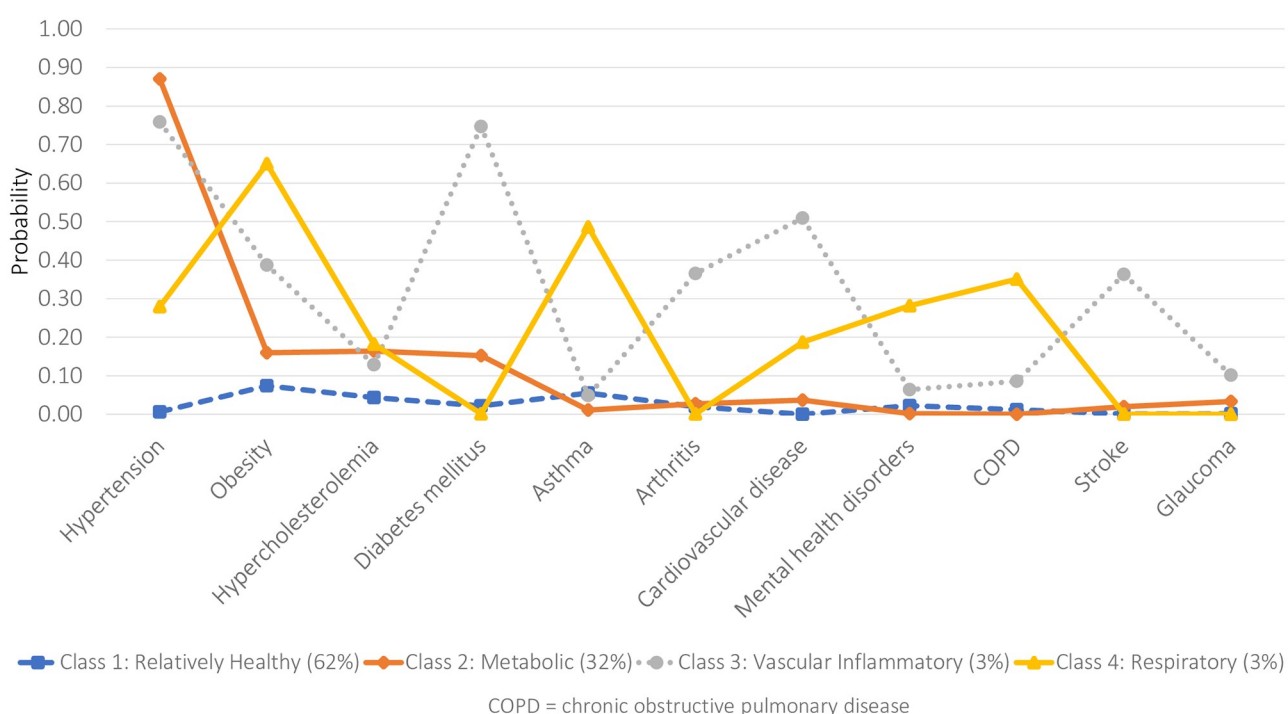

**Fig 3. Item-response probabilities for the four-class model (JHLS-II data, 2007/2008; Males only: n = 790).**

for each subsample, the alternative 5-class model performed no better than the 4-class one ($p_{\text{males}} = 0.38$; $p_{\text{females}} = 0.35$). The 4-class model was thus selected as the baseline model for optimal balance of model fit, parsimony and ease of interpretation.

For both males (Fig 3) and females (Fig 4), multimorbidity patterns generally mimicked the baseline model identified for the general population with differences, however, in both the prevalence of classes as well as the NCDs likely to be reported within each class.

Specifically, for males, the *Relatively Healthy* class comprised the majority of the sample (62.0%) and was characterized by individuals with an almost negligible probability of reporting any of the 11 NCDs. On the other hand, the *Relatively Healthy* class comprised just under half of the female sample (49.0%) and was characterized by individuals with a low probability of reporting any NCDs except obesity. The mean number of NCDs reported in this *Relatively Healthy* class was 0.3 and 0.6 for males and females, respectively. Almost equal proportions of the male and female subsamples (males: 32.1%; females: 31.8%) were classified into the second *Metabolic* class. Among males, however, hypertension was the only NCD of high probability while, among females, there was an increased likelihood of reporting hypertension and obesity. The mean number of NCDs reported in the *Metabolic* class was 1.4 and 1.9 for males and females, respectively. Only 3.0% of the male subsample was classified into the third *Vascular-Inflammatory* class, which was characterized by an increased probability of reporting hypertension, diabetes, cardiovascular disease, obesity, arthritis and stroke. In contrast, 14.7% of the female subsample was classified into the *Vascular-Inflammatory* class, which was characterized by an increased probability of reporting hypertension, obesity, arthritis, cardiovascular disease, hypercholesterolemia and diabetes. The mean number of NCDs reported in the *Vascular-Inflammatory* class was 3.8 and 3.6 for males and females, respectively. The final *Respiratory class* was characterized by individuals with a high probability of reporting obesity, asthma and

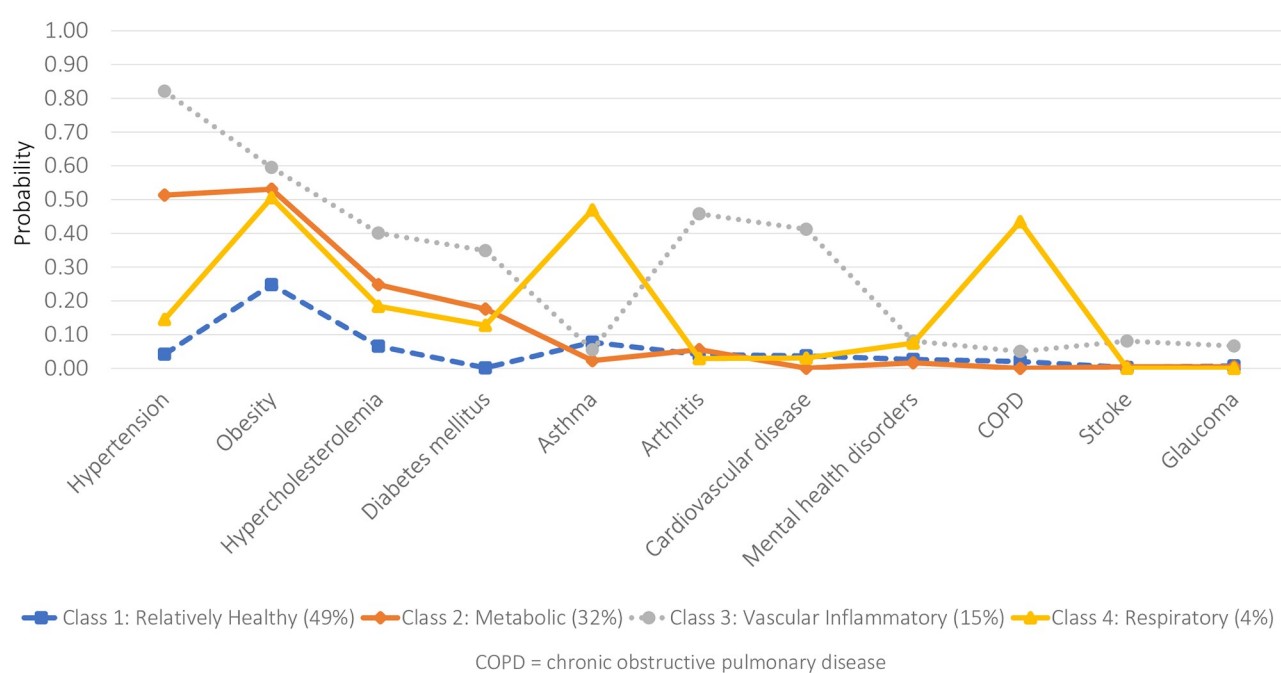

**Fig 4. Item-response probabilities for the four-class model (JHLS-II data, 2007/2008; Females only: n = 1,761).**

COPD, comprising 2.9% and 4.5% of the male and female subsamples, respectively. The mean number of NCDs reported was 3.3 and 2.7 for males and females, respectively.

### Exploratory factor analysis (EFA) results

Adequacy of the sample for factor analysis was confirmed by the KMO statistic of 0.7, which exceeded the recommended value of 0.6 [35,36]. Results supported evidence of three factors (i.e. multimorbidity patterns), with identification of three components with Eigenvalues greater than one. The scree plot showed the first major inflection (i.e. elbow) at the third factor, suggesting a similar retention of three factors for final analysis (Fig 5).

The three-factor solution collectively explained 60.3% of the variance of the total model, with each component explaining 29.7%, 19.3% and 11.3% of the variance, respectively. Following rotation, a simpler structure was identified with strong factor loadings on each of the three components, all having absolute values above the acceptable threshold of 0.3 (Table 5). Two NCDs (arthritis and cardiovascular disease) showed strong correlations with more than one factor (i.e. multimorbidity patterns).

Three multimorbidity patterns were identified in the Jamaican population using EFA: "vascular" (hypertension, obesity, hypercholesterolemia, diabetes, and stroke); "respiratory" (asthma and COPD), and "cardio-mental-articular" (cardiovascular disease, arthritis, and mental health disorders).

### Discussion

The burden of multimorbidity in the Caribbean has not yet been well described and this study is the first to use an LCA model to examine multimorbidity prevalence and patterns in the Jamaican population or the wider Caribbean region. Based on data on the presence or absence of 11 NCDs, four classes were identified, including a predominant *Relatively Healthy* class

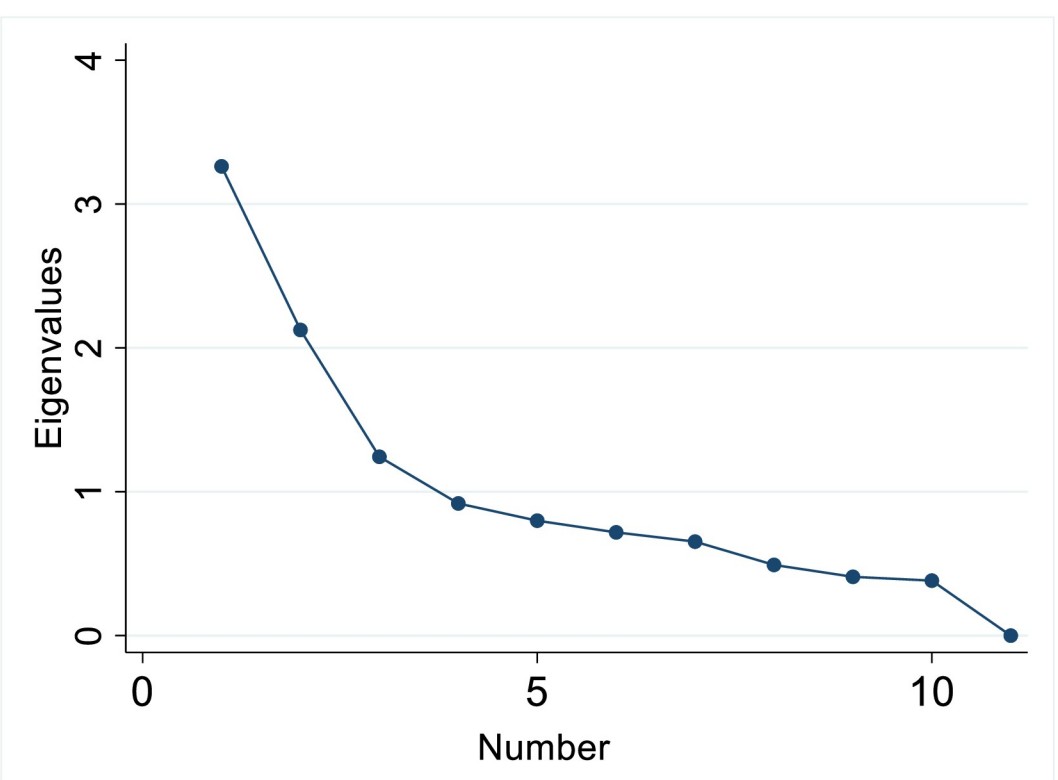

**Fig 5. Scree plot of eigenvalues (JHLS-II data, 2007/2008; N = 2,551).**

comprising 52.1% of the sample population and characterized by minimal disease. The other three classes were characterized by high burden of multimorbidity and, based on identified patterns, were labelled *Metabolic*, *Vascular-Inflammatory* and *Respiratory*. The resultant classes suggested an almost quantitative dimension to multimorbidity patterns (i.e. the average number of NCDs reported was higher with progressive classes), in addition to more distinct,

**Table 5. Factor scores and pattern matrix (JHLS-II data, 2007/2008; N = 2,551) for the 11 conditions of multimorbidity.**

|  | Factor 1: Vascular | Factor 2: Respiratory | Factor 3: Cardio-Mental-Articular |
|---|---|---|---|
| Hypertension | 0.76 |  |  |
| Obesity | 0.50 |  |  |
| Hypercholesterolemia | 0.61 |  |  |
| Diabetes | 0.71 |  |  |
| Asthma |  | 0.34 | -0.57 |
| Arthritis |  | 0.31 | 0.61 |
| Cardiovascular disease | 0.32 |  | 0.69 |
| Mental health disorders |  |  | 0.68 |
| COPD |  | 0.92 |  |
| Stroke | 0.79 |  |  |
| Glaucoma |  | 0.95 |  |

Extraction method: Principal component analysis. Rotation method: Oblique oblimin with Kaiser normalization.

qualitative differences in the types of diseases comprising the patterns (e.g. *Metabolic* vs. *Respiratory* classes).

Of note, the four diseases whose presence was determined using physical assessments (obesity, hypertension, diabetes, hypercholesterolemia) were primary contributors to multimorbidity patterns, particularly the *Metabolic* and *Vascular-Inflammatory* patterns. This may reflect greater certainty in objectively measured conditions. There was also a very high likelihood of reporting obesity across all multimorbidity classes. Sensitivity analyses, demonstrating that models with and without obesity were qualitatively similar, not only support the patterns identified but indicate the importance of obesity in increasing vulnerability to the accumulation of multiple chronic conditions in this population. This may explain the added vulnerability of women to the burden of multimorbidity, given that the prevalence of obesity among females is over three times as high as that in males. From a programmatic perspective, this finding also highlights the need to better target obesity, which has been identified as major public health problem throughout Jamaica [39,40], and the wider Caribbean region [41–43]. Indeed, while NCD prevention and control efforts should focus on addressing the complex needs of persons with multimorbidity, by supporting them to manage their existing conditions and prevent the accumulation of additional ones, activities need also focus on the *Relatively Healthy* subgroup for whom the presence of obesity may predispose to a multiplicity of chronic disorders. Evidence of high prevalence of obesity among Caribbean children and adolescents [41,44–46] further underscores the need for urgent intervention.

With regard to identified sex differences in this study population, findings suggest a similar structure in the overall patterning of multimorbidity among males and females, with some key differences in both the absolute burden of multimorbidity as well as the types of diseases comprising multimorbidity profiles among each sex. For example, while nearly two-thirds (62.0%) of the male sample was classified as *Relatively Healthy* with little probability of reporting any NCDs, the same was true for only about half (49.0%) of the female population. Further, the *Vascular-Inflammatory* class was considerably smaller among males (males = 3.0%; females = 14.7%) and was additionally characterized by a high likelihood of reporting stroke—suggesting that despite relatively low overall prevalence of this pattern, males with this disease profile may be at increased risk for complications, physical impairment and functional decline.

It is challenging to compare the results described here to findings from other studies, given differences in the number and type of disease indicators used to define multimorbidity, the types of populations sampled, and the statistical methods applied. Even among studies that have applied LCA to exploration of multimorbidity patterns, comparisons remain difficult since those studies were often limited to older population subgroups and included different disease spectra. Yet, among studies using LCA, results from this analysis were very similar to patterns identified in a population-based survey of Danish adults, aged 16 years and over, which identified seven classes with different disease patterns, based on 15 NCD indicators [9]. Specifically, comparable proportions of the samples (Jamaica$_{15-74 \text{ years}}$: 53% vs. Denmark$_{\geq 16 \text{ years}}$: 59%) were classified as *Relatively Healthy* with minimal probability of reporting any NCD, while the *Metabolic*, *Respiratory* and *Vascular-Inflammatory* classes identified in this study were qualitatively similar to the "Hypertension", "Complex Respiratory Disorders", "Complex Cardio-metabolic Disorders" classes, respectively, from the Danish study [9]. Although the Danish study identified three additional multimorbidity patterns [9], these disease profiles were likely not observed in the Jamaican sample since the presence/absence of diseases comprising these patterns (e.g. osteoporosis, slipped discs/other back injuries, migraine/recurrent headache, tinnitus, allergy) was not assessed in the JHLS-II survey.

In comparison to studies applying EFA to the exploration of multimorbidity, similarities in patterns are also observed. For example, one global study of multimorbidity patterns—using data on a cross-sectional sample of adults older than 50 years from the Collaborative Research on Ageing in Europe (COURAGE) project (in Finland, Poland, and Spain) as well as the WHO Study on Global Ageing and Adult Health (SAGE) survey (in China, Ghana, India, Mexico, Russia, and South Africa)–similarly observed a "Metabolic" (diabetes, obesity and hypertension) pattern of relevance to eight of the countries studied (i.e. China, Finland, Ghana, India, Poland, Russia, South Africa, Spain) as well as a "Respiratory" (asthma and COPD) pattern which was only relevant to two (i.e. Finland and Russia) [47]. Additional patterns, included a "cardio-respiratory" (i.e. angina, asthma, COPD) pattern of relevance to 7 of the countries studied (i.e. China, Ghana, India, Mexico, Poland, South Africa, Spain) and a "mental-articular" (i.e. arthritis, depression) pattern observed in 3 countries (i.e. China, Ghana, India) [47]. Evidence from high-income settings throughout Europe [6,9,34], North America [13,15], and Australia [11,12], have similarly demonstrated important differences in the type, prevalence and distribution of co-occurring conditions across populations. These results suggest that while clustering of diseases does exist, identification of context-specific multimorbidity patterns can enable better appreciation of disease burden and profiles, to meaningfully inform strategies aimed at prevention and control.

## Comparison of LCA vs. EFA

In this study, results from EFA were generally consistent with findings from LCA, with some minor differences. Both techniques identified three distinct *multimorbidity* patterns and suggested a prominence of two specific patterns of diseases (i.e. a respiratory pattern and a vascular pattern). The main difference was that in LCA there was a *Vascular-Inflammatory* class characterized by hypertension, obesity, hypercholesterolemia, diabetes, cardiovascular disease, and arthritis while, in EFA, the "vascular" pattern also included stroke but did not include arthritis. In fact, in EFA, a "cardio-mental-articular" factor emerged which included cardiovascular disease, arthritis, and mental health disorders. This factor was similar to the "mental-articular" (arthritis, depression) pattern described in above-mentioned global study [47], in addition to evidence from systematic reviews of multimorbidity patterns identified via EFA [7,48].

Observed differences between the EFA and LCA techniques may be attributed to the variable-centered approach of the former which is based on correlations between NCD indicators. Scientists have noted that EFA may also be problematic for binary data, which may be grouped owing to similar distributions rather than any common underlying features [18]. Conversely, the probabilistic LCA model uses a person-centered approach that may be more useful for strategic intervention planning by providing knowledge of the likelihood of individuals presenting with similar disease profiles. Indeed, LCA allowed for a more nuanced appreciation of two multimorbidity profiles—that is, a *Metabolic* class, with a strong likelihood of metabolic disorders only (e.g. hypertension, obesity) and another *Vascular-inflammatory* class where the probability of these two metabolic disorders was even higher and also coupled with increased likelihood of diabetes, hypercholesterolemia, arthritis and cardiovascular disease. This finding may suggest that the *Metabolic* subgroup is at risk of progression to a more severe *Vascular-Inflammatory* disease pattern where the burden of multimorbidity is higher. Although, empirical analyses indicate that those in the *Metabolic* group were significantly younger than those in the *Vascular-Inflammatory* group (mean age $_{Metabolic}$ = 46.1 vs. mean age $_{Vascular-Inflammatory}$ = 56.5; $p<0.001$), such a conclusion cannot be confirmed using the current study as longitudinal data is needed to explore risk of transitioning from one class to another.

## Strengths and limitations

This is the first study to assess profiles of co-occurrence of morbidities in Jamaica, or the larger Caribbean region. Via identification of distinct combinations, rather than simple counts of diseases, this study offers a richer and more nuanced understanding of multimorbidity prevalence and patterns in Jamaica, providing insight into the nature and severity of the NCD burden. It also adds to the evidence base of the multimorbidity burden in LMICs, providing data that is more comparable for other island nations which are similarly heavily affected by NCDs. Yet, there are several limitations. First, females outnumbered males nearly 2:1 in the final analytic sample and generalizability of results to the larger Jamaican population may be limited by the smaller proportion of males. Sex-specific LCA analyses may have also been limited by the small male subsample. Further, there is some degree of classification uncertainty in LCA [33] while different software packages have been noted to result in structurally different cluster solutions [49]. These limitations should be borne in mind in interpretation of results. Replication of our findings in future studies using larger samples would add further support for patterns identified.

Secondly, although use of both subjective self-reports and objective assessment may have increased reliability in measurement of four NCDs (hypertension, obesity, hypercholesterolemia, diabetes), accuracy in reporting of the other seven NCDs may have been affected by various factors. For example, inaccurate self-reporting of prevalent mental health disorders is noted in the literature [50] and may reflect diseases being undiagnosed or failure of participants to disclose their conditions to interviewers, while diseases such as asthma tend to be more commonly diagnosed in children and youth [51]. Another limitation is that this study was unable to assess either disease severity or the presence/absence of pain, both of which may not only influence participant self-reports but also serve as important indicators of disease control and individual capacity. The decision to include self-reported bronchitis/pneumonia as a proxy for COPD may not be supported by other researchers who may query inclusion of this disease type within the NCD umbrella. Further, given that the final list of 11 conditions was based largely on convenience and limited to those NCDs identified in the JHLS-II survey questionnaire, it is likely that different multimorbidity profiles may have emerged if other NCD indicators had been used. Notably, however, in the absence of a gold standard measure for multimorbidity, adherence to recommended standards, which advise inclusion of between 11–12 most prevalent chronic diseases in a given population [4,10], is a major strength of this study. This study included all diseases specified in the recommended list, with the exception of cancer—given its lower overall prevalence in the sample population.

Finally, while use of population-level data increased the representativeness of identified patterns, the study design which excluded age-groups older than 74 years may have introduced a selection and information bias. It is well-recognized that multimorbidity assumes greater importance with advancing age [4,7,10] and failure to examine variations in patterns as people age omits an important population demographic where multimorbidity may be more common, with greater implications for disease severity, management of conditions, functional status and quality of life.

## Conclusion

The findings indicate that a considerable proportion of the population is managing two or more conditions, with a female preponderance in the burden and degree of multimorbidity. Consistency of multimorbidity patterns identified here with results from other international studies supports the non-random association of diseases and the need for intervention to better control and support, if not prevent, the inevitable lifelong management of multiple diseases

with which many populations must contend. Future work using longitudinal datasets would enable exploration of disease trajectories and understanding of how individuals manage multiple conditions and transition to different patterns over time. Investigation of multimorbidity burden in other LMICs is also needed to better reflect individual burden of disease as well as clinician's daily workload and experience. As future research continues to examine this multimorbidity phenomenon, exploration into the causes and consequences of NCD patterns, with attention to variation in disease profiles according to sex, age and socio-economic status, can guide the development of strategies that allow for more targeted prevention and intervention.

## Supporting information

**S1 Data. Data underlying the present analyses.**
(XLS)

**S1 Table. Model comparison for selecting number of multimorbidity latent classes based on 10 indicators (JHLS-II data, 2007/2008; N = 2,551).** This LCA model excludes obesity.
(PDF)

**S1 Fig. Item-response probabilities for the latent class model based on 10 indicators (JHLS-II data, 2007/2008; N = 2,551).** This LCA model excludes obesity.
(TIF)

## Acknowledgments

We would like to acknowledge the contribution of Prof. Rainford Wilks and Dr. Novie Younger-Coleman who granted us access to the JHLS-II data and provided intellectual and statistical support in the analysis of data and writing of this manuscript.

## Author Contributions

**Conceptualization:** Leslie S. Craig.

**Formal analysis:** Leslie S. Craig.

**Writing – original draft:** Leslie S. Craig.

**Writing – review & editing:** Leslie S. Craig, David R. Hotchkiss, Katherine P. Theall, Colette Cunningham-Myrie, Julie H. Hernandez, Jeanette Gustat.

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
