## [Decision Letter · Decision Letter 0]

6 Apr 2020

PONE-D-20-03727

Patterns of multimorbidity in the Jamaican population: A comparative analysis of latent variable models

PLOS ONE

Dear Dr. Cunningham-Myrie,

Thank you for submitting your manuscript to PLOS ONE. After careful consideration, we feel that it has merit but does not fully meet PLOS ONE’s publication criteria as it currently stands. Therefore, we invite you to submit a revised version of the manuscript that addresses the points raised during the review process.

We would appreciate receiving your revised manuscript by May 21 2020 11:59PM. To enhance the reproducibility of your results, we recommend that if applicable you deposit your laboratory protocols in protocols.io, where a protocol can be assigned its own identifier (DOI) such that it can be cited independently in the future. For instructions see: http://journals.plos.org/plosone/s/submission-guidelines#loc-laboratory-protocols

We look forward to receiving your revised manuscript.

Kind regards,

Khin Thet Wai, MBBS, MPH, MA (Population & Family Planning Resear

Academic Editor

PLOS ONE

Additional Editor Comments (if provided):

This is the interesting research work highlighting the multimorbidity pattern in a developing region that would draw attention for changes in resources allocation and supporting strategies for health planning.

According to reviewers, the statistical approach (LCA and EFA) to analyze the data requires a revision to strengthen the scientific integrity in data interpretation and valid conclusions. In addition, more subjective argument is critical .

Reviewers' comments:

Reviewer's Responses to Questions

**Comments to the Author**

1. Is the manuscript technically sound, and do the data support the conclusions?

Reviewer #1: Yes

Reviewer #2: Partly

Reviewer #3: Partly

2. Has the statistical analysis been performed appropriately and rigorously? 

Reviewer #1: No

Reviewer #2: Yes

Reviewer #3: No

3. Have the authors made all data underlying the findings in their manuscript fully available?

Reviewer #1: Yes

Reviewer #2: Yes

Reviewer #3: Yes

4. Is the manuscript presented in an intelligible fashion and written in standard English?

Reviewer #1: Yes

Reviewer #2: Yes

Reviewer #3: Yes

5. Review Comments to the Author

Reviewer #1: General remark: This article examines the patterns of multimorbidity among the adult population of Jamaica. The authors use advanced statistical techniques to analyze data and the paper is well written. I find the study interesting, but there are some methodological weaknesses in the study that I think should be addressed.

Page 8, line 147 ff.

The following description in the manuscript of the sample weighting procedure could leave the expression that the weighted sample is representative of the Jamaican population:

"The final analytic sample of 2,551 respondents included 790 males and 1,761 females … All analyses were weighted to account for sampling design and non-response as well as differences in the age-sex distribution of the study sample compared to the Jamaican population. Base sampling weights reflected the product of the inverse of the probability of selecting a household and the inverse of the selecting a primary sampling unit, adjusted for non-response. Post-stratification weights were calculated as the number of persons in the Jamaican population between the ages of 15-74 years represented by each individual in the sample within 5-year age-sex categories."

However, when this reviewer calculated the proportion of males using the attached data file, the unweighted percentage of males was 30.97 while the weighted percentage was 31.55 (using the variable sampwt_adj as weight). This indicates that the weighted results are far from representative of the Jamaican adult population (49% of the population were men according to Wilks R, Younger N, Tulloch-reid M, Mcfarlane S, Francis D (2008) Jamaica Health and Lifestyle Survey 2007-8. Kingston: Tropical Medicine Research Institute, University of the West Indies). This could potentially have far-reaching consequences for the quality of the study. The authors should comment on this. Is it possible to generalize the results to the entire population?

Page 8, line 214

"All statistical analyses were carried out via Stata v.15 software, using the LCA Stata Plugin …"

It is well-known that different LCA-programs may come up with different cluster solutions (cf. Haughton, Dominique & Legrand, Pascal & Woolford, Sam. (2009). Review of three latent class cluster analysis packages: Latent gold, poLCA, and MCLUST. The American Statistician. 63. 81-91. 10.1198/tast.2009.0016.)

The most widely used LCA programs are LatentGOLD and MPlus. I tried to repeat the LCA analysis on page 12 using LatentGOLD (without and with weights). The results differ from the results in Table 1, see below:

File name:S1 Data.sav

File size:2551 records

File date:2020-feb-610:06:55

LLBIC(LL)AIC(LL)NparL²dfp-valueClass.Err.

UNWEIGHTED

Model11-Cluster-23246924649533464940711374390,120362,9e-785550

Model22-Cluster-22223704445051444478723169745,820243,4e-344780,0997

Model33-Cluster-22119714424414442401235148947,520121,7e-300300,1956

Model44-Cluster-22037994408232440769147132602,720006,3e-265430,2154

Model55-Cluster-2199355439950643988285912371519887,7e-246540,2047

Model66-Cluster-2195780439251843917027111656519761,6e-231370,2345

Model77-Cluster-21937114388541438758783112426,719645,8e-222650,3091

WEIGHTED

Model81-Cluster-8247,8116581,9216517,63111315,148203610

Model92-Cluster-7876,2315932,8915798,4723571,9874202410,1051

Model103-Cluster-7836,3815947,3115742,7635492,2788201210,1635

Model114-Cluster-7810,715990,0815715,447440,9214200010,221

Model125-Cluster-7797,7316058,2715713,4659414,9783198810,2792

Model136-Cluster-7786,2116129,3615714,4171391,9334197610,279

Model147-Cluster-7775,1516201,3715716,383369,8185196410,2633

In the example with unweighted data, BIC and AIC did not reach a minimum among models with 1-7 latent classes. In the example with the weighted data, a minimum is reached for BIC with a three-cluster solution and for AIC with a five-cluster solution. Compared to this, in Table 1, models with more than four classes are not well identified. I suggest that the authors consider running the analysis again with another LCA program to ensure that results are stable.

Reviewer #2: Craig et al. intended to estimate the prevalence of multimorbidity, identify population subgroups with similar disease profiles, and examine consistency in patterns with different statistical approaches. They analyzed 2,551 subjects and observed a high burden of co-existing conditions that is predominantly borne by female.

1.Line 152 Pearson “?”. What statistic? Pearson correlation statistics? Spell out the full name. if it’s Pearson correlation, then please explain how it can be used to examine differences across sex as stated in the manuscript.

2. Line 307, “in addition to examination of the distribution of item-response probabilities across all solutions, the 4-class model appeared to provide the best interpretability in each case.” As the discrepancy was observed from different criteria, it requires more explanation how this conclusion was reached. The argument based on the interpretation is somewhat rather subjective!

3.In the exploratory factor analysis, three factors were identified. The LCA, four were identified. Why did authors conclude the consistent results among different approach?

4.The EFA was only performed for all sample. The comparison for the sex-specific analysis should be conduct as the discrepancies were identified earlier.

Reviewer #3: This in an interesting paper, however I have few comments to authors as follows.

Although the title of the paper indicates the paper focused on identifying patterns of multimorbidity, the objectives includes measure the prevalence, identifying the patterns of multimorbidity and comparing between the methods. Including multiple objectives has prevented the authors to produce rigorous results and conclusion for the paper. I would recommend the author keep the main objective in according to the title of the paper. The author should decide which methods (LCA or EFA) as the main approach used to identify the pattern of multimorbidity of course with the justification for the decision, the other can be used as a robustness check for their results.

While the authors provided reference to link with other source where study designs and data collection were described, the method of this paper does not have information on study design, data collection and study population to justify for whether the data is sufficient to address the aim of this study. I think the author should include the information in the paper.

Lines 145-147: “There were no statistically significant differences 146 between those with complete and those with missing information on the basis on sex, age or region of 147 residence (data not shown)”. How about the difference between the study population and the target population whether the sampling frame used?

The two methods were used to identify the patterns of multimorbidity, however, I don’t find any justification for why the authors used the method. What are the contribution of the methods in identifying the patterns? If you intend to compare both methods to see which one is better to use, the simulated data with a known pattern of multimorbidity should be used instead of the current dataset where the true pattern is unknown.

Four classes of multimorbidity were identified in the paper, it would be more informative if the authors have further details of what are the characteristics of the classes such as age, social economic status etc. Whether the characteristics are significant different across the classes of multimorbidity.

6. PLOS authors have the option to publish the peer review history of their article (what does this mean?). If published, this will include your full peer review and any attached files.

Reviewer #1: No

Reviewer #2: No

Reviewer #3: No

---

## [Author Response · Author response to Decision Letter 0]

15 May 2020

Reviewer #1

General remark: This article examines the patterns of multimorbidity among the adult population of Jamaica. The authors use advanced statistical techniques to analyze data and the paper is well written. I find the study interesting, but there are some methodological weaknesses in the study that I think should be addressed.

Page 8, line 147 ff.

The following description in the manuscript of the sample weighting procedure could leave the impression that the weighted sample is representative of the Jamaican population:

"The final analytic sample of 2,551 respondents included 790 males and 1,761 females. All analyses were weighted to account for sampling design and non-response as well as differences in the age-sex distribution of the study sample compared to the Jamaican population. Base sampling weights reflected the product of the inverse of the probability of selecting a household and the inverse of the selecting a primary sampling unit, adjusted for non-response. Post-stratification weights were calculated as the number of persons in the Jamaican population between the ages of 15-74 years represented by each individual in the sample within 5-year age-sex categories."

However, when this reviewer calculated the proportion of males using the attached data file, the unweighted percentage of males was 30.97 while the weighted percentage was 31.55 (using the variable sampwt_adj as weight). This indicates that the weighted results are far from representative of the Jamaican adult population (49% of the population were men according to Wilks R, Younger N, Tulloch-Reid M, Mcfarlane S, Francis D (2008) Jamaica Health and Lifestyle Survey 2007-8. Kingston: Tropical Medicine Research Institute, University of the West Indies). This could potentially have far-reaching consequences for the quality of the study. The authors should comment on this. Is it possible to generalize the results to the entire population?

Response:

This is an important point. The reviewer is correct that based on the (un)weighted percentage of males (~31%) in the sample, females outnumber males nearly 2:1. Of note, the JHLS-II study investigators created sampling weights, including post-stratification weights, that attempt to compensate for sampling inadequacies and, when used, yield age-specific proportions which are in keeping with national statistics (Wilks, Younger, Tulloch-Reid, Mcfarlane, & Francis, 2008). This adds support that the data remain nationally representative. Nonetheless, we agree that potential limitations regarding generalizability of results should be acknowledged and have done so in the Discussion section.

Page 8, line 214

"All statistical analyses were carried out via Stata v.15 software, using the LCA Stata Plugin."

It is well-known that different LCA-programs may come up with different cluster solutions (cf. Haughton, Dominique & Legrand, Pascal & Woolford, Sam. (2009). Review of three latent class cluster analysis packages: Latent gold, poLCA, and MCLUST. The American Statistician. 63. 81-91. 10.1198/tast.2009.0016.) The most widely used LCA programs are LatentGOLD and MPlus. I tried to repeat the LCA analysis on page 12 using LatentGOLD (without and with weights). The results differ from the results in Table 1, see below:

File name: S1 Data.sav 

File size: 2551 records 

File date: 2020-feb-6 10:06:55 

 LL BIC(LL) AIC(LL) Npar L² df p-value Class.Err.

UNWEIGHTED

Model1 1-Cluster -2324692 4649533 4649407 11 374390,1 2036 2,9e-78555 0

Model2 2-Cluster -2222370 4445051 4444787 23 169745,8 2024 3,4e-34478 0,0997

Model3 3-Cluster -2211971 4424414 4424012 35 148947,5 2012 1,7e-30030 0,1956

Model4 4-Cluster -2203799 4408232 4407691 47 132602,7 2000 6,3e-26543 0,2154

Model5 5-Cluster -2199355 4399506 4398828 59 123715 1988 7,7e-24654 0,2047

Model6 6-Cluster -2195780 4392518 4391702 71 116565 1976 1,6e-23137 0,2345

Model7 7-Cluster -2193711 4388541 4387587 83 112426,7 1964 5,8e-22265 0,3091

WEIGHTED 

Model8 1-Cluster -8247,81 16581,92 16517,63 11 1315,148 2036 1 0

Model9 2-Cluster -7876,23 15932,89 15798,47 23 571,9874 2024 1 0,1051

Model10 3-Cluster -7836,38 15947,31 15742,76 35 492,2788 2012 1 0,1635

Model11 4-Cluster -7810,7 15990,08 15715,4 47 440,9214 2000 1 0,221

Model12 5-Cluster -7797,73 16058,27 15713,46 59 414,9783 1988 1 0,2792

Model13 6-Cluster -7786,21 16129,36 15714,41 71 391,9334 1976 1 0,279

Model14 7-Cluster -7775,15 16201,37 15716,3 83 369,8185 1964 1 0,2633

In the example with unweighted data, BIC and AIC did not reach a minimum among models with 1-7 latent classes. In the example with the weighted data, a minimum is reached for BIC with a three-cluster solution and for AIC with a five-cluster solution. Compared to this, in Table 1, models with more than four classes are not well identified. I suggest that the authors consider running the analysis again with another LCA program to ensure that results are stable.

Response:

This is another excellent point. Unfortunately, we do not have access to any of the LCA programs you note (i.e., LatentGOLD, MPlus). We do, however, have access to SAS software and have repeated analyses using this statistical package. Results are presented below:

UNWEIGHTED 

# of Classes G2 df AIC BIC Adjusted BIC log-likelihood

1 1315.15 2036 1337.15 1401.43 1366.48 -8247.81

2 571.98 2024 617.98 752.40 679.32 -7876.23

3 498.37 2012 568.37 772.92 661.71 -7839.42

4 440.86 2000 534.86 809.54 660.21 -7810.67

5 NOT WELL IDENTIFIED

6 395.27 1976 537.27 952.21 726.63 -7787.88

WEIGHTED 

# Latent Classes G2 df AIC BIC Adjusted BIC log-likelihood

1 1318.43 2036 1340.43 1404.71 1369.76 -8186.47

2 597.76 2024 643.76 778.18 705.10 -7826.14

3 524.52 2012 594.52 799.07 687.87 -7789.52

4 466.96 2000 560.96 835.64 686.31 -7760.74

5 435.66 1988 553.66 898.47 711.01 -7745.09

6 410.48 1976 552.48 967.42 741.84 -7732.5

As is the case with your weighted example and the results presented in the manuscript, the BIC reached a minimum with the 2-class solution for both unweighted and weighted analyses in SAS. As noted in the manuscript, however, while we considered information criteria together with substantive interpretability (i.e., that resultant solutions were distinguishable, non-trivial in size, and meaningful) to determine the best fit solution, greater weight was given to the AIC and adjusted BIC following evidence from simulation studies of serious underfitting of the BIC, particularly with smaller samples and more unequal class sizes (Dziak, Coffman, Lanza, Li, & Jermiin, 2019; Nylund, Asparouhov, & Muthén, 2007).

For the unweighted SAS analyses, both the AIC and adjusted BIC reach a minimum with the 4-class model. This is in keeping with the results presented in the manuscript. Also, the 5-class model is similarly not well identified in the SAS example (i.e. the model did not converge in 5,000 iterations) and, while the 6-class model is well-identified in the SAS model, none of the information criteria indicate better fit over the 4-class solution. Differences in model identification for analyses performed in SAS and Stata (and perhaps in the LatentGOLD results the reviewer provides above) may reflect Stata not exceeding a maximum set of 400 iterations. 

For the weighted SAS analyses, the AIC and adjusted BIC suggested 6-class and 4-class solutions, respectively. Of note, however, the 6-class solution did not meet criteria for substantive interpretability (i.e., classes were neither distinguishable, non-trivial in size, nor meaningful). Results of the 6-class solution in SAS are presented below to illustrate this point:

Class Prevalences 1 2 3 4 5 6

 0.47 0.23 0.10 0.16 0.02 0.02

Item-response probabilities 1 2 3 4 5 6

Hypertension 0.01 0.51 0.88 0.59 0.24 0.10

Obesity 0.18 0.68 0.57 0.00 0.26 0.50

Hypercholesterolemia 0.05 0.23 0.42 0.16 0.00 0.24

Diabetes 0.00 0.15 0.42 0.14 0.08 0.06

Asthma 0.08 0.06 0.05 0.00 0.08 0.43

Arthritis 0.03 0.10 0.41 0.00 0.23 0.02

CVD 0.00 0.00 0.42 0.00 1.00 0.12

Mental health disorders 0.03 0.02 0.08 0.01 0.07 0.13

COPD 0.02 0.00 0.06 0.00 0.03 1.00

Stroke 0.00 0.00 0.12 0.02 0.00 0.00

Glaucoma 0.00 0.00 0.07 0.04 0.07 0.00

Please note also that weighted and unweighted analyses in Stata yielded qualitatively similar results – that is, identifying the 4-class model comprising Relatively Healthy, Metabolic, Vascular-Inflammatory and Respiratory classes as the best fit model. 

Nonetheless, we thank the reviewer for this comment as the concern raised regarding different LCA programs leading to different solutions is an important one. We have added mention of this in the Discussion section, including reference to the Haughton et al (2009) article kindly suggested by the reviewer, noting potential limitations of our analyses and implications for future work applying LCA techniques.

Reviewer #2

Craig et al. intended to estimate the prevalence of multimorbidity, identify population subgroups with similar disease profiles, and examine consistency in patterns with different statistical approaches. They analyzed 2,551 subjects and observed a high burden of co-existing conditions that is predominantly borne by females.

1.Line 152 Pearson "?". What statistic? Pearson correlation statistics? Spell out the full name. if it's Pearson correlation, then please explain how it can be used to examine differences across sex as stated in the manuscript.

Response:

Thank you for this comment. This should have read “Pearson's chi-squared (ꭓ2) test”. This test was used to examine differences in the prevalence of single and multiple morbidities across sex. It has been corrected in the text.

2. Line 307, "in addition to examination of the distribution of item-response probabilities across all solutions, the 4-class model appeared to provide the best interpretability in each case." As the discrepancy was observed from different criteria, it requires more explanation how this conclusion was reached. The argument based on the interpretation is somewhat rather subjective!

Response:

We are thankful to the reviewer for making this important point. Indeed, the selection of the best fit solution for the male and female subsamples were not as clear based on information criteria alone. Specifically, in the sex-specific analyses, “… the AIC suggested a 4-class model while the BIC suggested the 2-class model. However, for the male cohort, the adjusted BIC reached a minimum with the 3-class model while, for the female cohort, it did so with the 2-class model.”

As noted in the manuscript, we did give greater weight to the AIC and the adjusted BIC following evidence from simulation studies of serious underfitting of the BIC, particularly with smaller samples and more unequal class sizes (Dziak et al., 2019; Nylund et al., 2007). Further, as guided by recommendations from Collins and Lanza (Collins & Lanza, 2010) on determination of the best fit solution, we ensured that our final decision-making was informed by thorough evaluation of a wide range of available information. This included examination of substantive interpretability (i.e., ensuring that resultant solutions were distinguishable, non-trivial in size, and meaningful), comparison of entropy scores (to reflect classification certainty and model precision) and use of parametric bootstrap likelihood ratio tests (to test the null hypothesis that the specified LCA model fit the data) (S. Lanza & Rhoades, 2013; S. T. Lanza, Collins, Lemmon, & Schafer, 2007). Collectively, these criteria and parameters gave greater support for the 4-class model. Yet, the reviewer is correct that this final determination is somewhat subjective. In light of this important point, we have added a sentence to the Discussion section acknowledging the need for replication in future work with larger samples.

3.In the exploratory factor analysis, three factors were identified. The LCA, four were identified. Why did authors conclude the consistent results among different approach?

Response: 

This is an excellent point. The number of patterns (i.e. classes vs. factors) observed speak to the differences in the two statistical approaches. Both techniques identified three multimorbidity patterns. LCA revealed three multimorbidity profiles in addition to a no disease or single morbidity profile (i.e. 4 classes in total). EFA revealed 3 multimorbidity factors – just as LCA did. The EFA approach would not have identified a no disease or single morbidity factor since strong, stable factors typically contain at least three strongly loading items (i.e., items with factor loadings ≥0.5) (Costello & Osbourne, 2005). (Of note, in this study, we specified that 2 items were sufficient for identification of a factor given the current definition of multimorbidity as the simultaneous presence of two or more diseases.)

4.The EFA was only performed for all sample. The comparison for the sex-specific analysis should be conduct as the discrepancies were identified earlier.

Response: 

Thank you, this is another excellent point. We chose not to present sex-specific EFA analyses given the many results already presented in the paper, along with previously acknowledged sample size limitations (i.e., the small male subsample) and cautioning that EFA is a “large-sample” procedure (Costello & Osbourne, 2005). Yet, results from sex-specific analyses were qualitatively similar to EFA results on the full sample, identifying “vascular”; “respiratory”, and “cardio-mental-articular” factors. Results were also consistent with LCA subgroup analyses. Specifically, among men, the vascular factor (which was qualitatively similar to the Vascular-Inflammatory class) included hypertension, diabetes, cardiovascular disease, obesity and stroke while, among females, this factor included hypertension, obesity, hypercholesterolemia and diabetes. 

 Men Women

 Vascular Respiratory Cardio-Mental-Articular Vascular Respiratory Cardio-Mental-Articular

Hypertension 0.62 0.81 

Obesity 0.60 0.50 

Hypercholesterolemia 0.70 0.63 

Diabetes 0.77 0.70 

Asthma 0.94 0.33 -0.50

Arthritis 0.77 0.41 0.61

CVD 0.77 0.38 0.68

Mental health disorders -0.88 0.64

COPD -0.39 0.55 0.46 0.79 -0.31

Stroke 0.74 0.47 0.64 -0.71 

Glaucoma 0.76 0.35 0.96 

Reviewer #3

This in an interesting paper, however I have few comments to the authors as follows.

Although the title of the paper indicates the paper focused on identifying patterns of multimorbidity, the objectives include measuring the prevalence, identifying the patterns of multimorbidity and comparing between the methods. Including multiple objectives has prevented the authors to produce rigorous results and conclusion for the paper. I would recommend the author keep the main objective in, according to the title of the paper. The author should decide which methods (LCA or EFA) is the main approach used to identify the pattern of multimorbidity, of course with the justification for the decision; the other can be used as a robustness check for their results.

Response:

This is an important point and we appreciate the reviewer’s comments. The title has been revised to “Prevalence and patterns of multimorbidity in the Jamaican population: A comparative analysis of latent variable models”. We believe this title better captures the aims which the paper attempts to address. 

We have also revised the paper throughout to better reflect the goals of this manuscript in describing the prevalence and patterns of multimorbidity in the Jamaican population, using LCA as the main modelling approach, with EFA used to examine whether results are robust to variation in how multimorbidity patterns are analyzed.

While the authors provided a reference to link with another source where study designs and data collection were described, the method of this paper does not have information on study design, data collection and study population to justify whether the data is sufficient to address the aim of this study. I think the author should include the information in the paper.

Response:

Thank you for this important comment. In the revised manuscript, we now provide more details on the study design and sampling strategy for this nationally representative survey.

Lines 145-147: "There were no statistically significant differences 146 between those with complete and those with missing information on the basis on sex, age or region of 147 residence (data not shown)". How about the difference between the study population and the target population whether the sampling frame used?

Response:

This is an important point and we thank the reviewer for this comment. As a secondary data analysis, we do not have access to the sampling frame used; however, we do acknowledge that previous researchers have noted that errors in the sampling frame are possible and may account for the excess of females in the sample as well as lower representation of some age-groups than would be expected based on population statistics (Tulloch-Reid et al., 2013; Wilks et al., 2008). Notably, however, post-stratification weights were designed by the JHLS-II study investigators to account for sampling inadequacies and, when used, yield age-specific proportions that are in keeping with national statistics and add support for the comparability and representativeness of data (Wilks et al., 2008). Revisions to the Discussion section address important considerations for the representativeness of study findings.

The two methods were used to identify the patterns of multimorbidity, however, I don't find any justification for why the authors used the method. What are the contribution of the methods in identifying the patterns? If you intend to compare both methods to see which one is better to use, the simulated data with a known pattern of multimorbidity should be used instead of the current dataset where the true pattern is unknown.

Response:

Thank you for this comment. The choice to use LCA was based on its person-centered approach, combined with evidence of its increasing application to studies on multimorbidity patterns. However, among identified gaps and next steps for multimorbidity research, scientists have consistently called for context-specific multimorbidity studies that also “increase the reliability of findings through comparison of statistical techniques.” We thus chose to compare LCA results with those from EFA since the latter method is also a latent modelling technique and has been cited as one of the commonly used methods (aside from simple counts) for examining multimorbidity patterns. The reviewer makes a great point with regard to true patterns and simulated data. As the first study, to our knowledge, to examine patterns of multimorbidity in a Caribbean population, true patterns were unfortunately unknown at the time of this research. We have thus revised the text to better clarify the additional use of EFA, not merely as a comparison technique but also as a robustness check. We have also revised the manuscript to note that future replication studies could provide additional support for patterns identified.

Four classes of multimorbidity were identified in the paper, it would be more informative if the authors have further details of what are the characteristics of the classes such as age, social economic status etc. Whether the characteristics are significantly different across the classes of multimorbidity.

Response:

This is an excellent point and the authors agree that information on the social profiles of multimorbidity would be very informative. Unfortunately, given the many aims described in this paper, an in-depth exploration of social determinants was beyond the scope of this manuscript. However, further analyses describing the social determinants of the identified patterns are underway.

---

## [Decision Letter · Decision Letter 1]

29 Jun 2020

Prevalence and patterns of multimorbidity in the Jamaican population: A comparative analysis of latent variable models

PONE-D-20-03727R1

Dear Dr. Cunningham-Myrie,

We’re pleased to inform you that your manuscript has been judged scientifically suitable for publication and will be formally accepted for publication once it meets all outstanding technical requirements.

Kind regards,

Khin Thet Wai, MBBS, MPH, MA (Population & Family Planning Resear

Academic Editor

PLOS ONE

Additional Editor Comments (optional):

Reviewers' comments:

Reviewer's Responses to Questions

**Comments to the Author**

1. If the authors have adequately addressed your comments raised in a previous round of review and you feel that this manuscript is now acceptable for publication, you may indicate that here to bypass the “Comments to the Author” section, enter your conflict of interest statement in the “Confidential to Editor” section, and submit your "Accept" recommendation.

Reviewer #1: (No Response)

Reviewer #2: All comments have been addressed

Reviewer #3: All comments have been addressed

2. Is the manuscript technically sound, and do the data support the conclusions?

Reviewer #1: Yes

Reviewer #2: (No Response)

Reviewer #3: Yes

3. Has the statistical analysis been performed appropriately and rigorously? 

Reviewer #1: Yes

Reviewer #2: (No Response)

Reviewer #3: Yes

4. Have the authors made all data underlying the findings in their manuscript fully available?

Reviewer #1: Yes

Reviewer #2: (No Response)

Reviewer #3: Yes

5. Is the manuscript presented in an intelligible fashion and written in standard English?

Reviewer #1: Yes

Reviewer #2: (No Response)

Reviewer #3: Yes

7. PLOS authors have the option to publish the peer review history of their article (what does this mean?). If published, this will include your full peer review and any attached files.

Reviewer #1: **Yes: **Finn Breinholt Larsen

Reviewer #2: No

Reviewer #3: No

6. Review Comments to the Author

Reviewer #2: (No Response)

Reviewer #3: (No Response)

---

## [Editor Report · Acceptance letter]

9 Jul 2020

PONE-D-20-03727R1 

Prevalence and patterns of multimorbidity in the Jamaican population: A comparative analysis of latent variable models 

Dear Dr. Cunningham-Myrie:

I'm pleased to inform you that your manuscript has been deemed suitable for publication in PLOS ONE. Congratulations! Your manuscript is now with our production department. 

Kind regards, 

on behalf of

Dr. Khin Thet Wai 

Academic Editor

PLOS ONE